# Review of the Impact of IT on the Environment and Solution with a Detailed Assessment of the Associated Gray Literature

**Bourgeois Guillaume** [1,*] **, Duthil Benjamin** [1,2] **and Courboulay Vincent** [1]

1    IT, Image, Interaction Laboratory (L3I), University of la Rochelle, 17000 La Rochelle, France;
     duthil@eigsi.fr (D.B.); vcourbou@univ-lr.fr (C.V.)
2    EIGSI, 17041 La Rochelle, France
*    Correspondence: guillaume.bourgeois@univ-lr.fr

**Abstract:** The immaterial aspect of digital technology tends to make us forget its growing impact on the environment. Today, the situation has changed: we are becoming aware of the material aspect of digital technology, especially with the recent datacenter fires. The topic is now on the political agenda, and for good reason: digital accounts for nearly 4% of global emissions, and emissions from the sector are growing exponentially. Digital impact analysis is crucial to increase the visibility and consequently the diffusion and democratization of responsible digital. This article offers a detailed study and solutions on the environmental impact of IT with a detailed assessment of the gray literature of the last years and accompanying elements such as LCA or ISO standards. Thus, we have a view of the main green IT tools, the environmental impact of digital on datacenters, user equipment and networks, recent forecasts, and a look at the future challenges of digital technologies (such as AI or Blockchain),and finally a conclusion with the limitations of our research.

**Keywords:** green IT; sustainable information systems; literature review; environmental awareness; green computing; ecological impact; sustainable development; energy consumption; solutions; gray literature





## 1. Introduction

It is within a global framework that the Paris Agreement limits the increase in global temperature to "well below 2 °C" by 2100 and urges states to continue their efforts to reach +1.5 °C. To achieve this ambitious goal, the agreement calls for "a balance between human-induced emissions and the Earth's natural absorptive capacity, to ensure that carbon sinks such as forests play a role" [1]. While the text does not mention quantitative reductions in greenhouse gas emissions, the IPCC (Intergovernmental Panel on Climate Change) indicates that in order to keep global warming below 1.5 degrees, greenhouse gas emissions must be reduced by 70–80% over half a century. The goal of zero emissions will be reached by 2100 at the latest.

At the same time, most households and businesses use information and communication technologies (ICT) on a daily basis. These technologies include computers, telecommunications and electronics. Attention to the environment is therefore inevitable. These technologies are generally considered as "immaterial" and their impact on the planetary ecology seems to be ignored. They convey images of lightness [2]. On the other hand, the energy intensity of the digital industry is growing at a rate of 4% per year: it runs counter to the trend of the energy intensity of the global GDP, which is currently decreasing at a rate of 1.8% per year. The rise of video usage (Skype, streaming, etc.) and the increase in the number of frequently replaced digital devices are the main drivers of the expansion in energy consumption [2,3].

Thus, since the 1980s, the number of functional computer equipment in the world has not stopped increasing. This is why "ecological ICT" or "green ICT" was born, defined

as "information and communication technologies" by the General Commission for Terminology and Neology, which allow, through the design or use of means of communication, a reduction of the negative impact of human activities on the environment". In summary, the concept of responsible digital represents a series of measures taken by the industry to reduce the ecological, economic and social footprint of ICT. It is a comprehensive and coherent method to reduce the problems encountered mainly in the design, use and end of life of IT equipment [4].

Thus, we also talk a lot about ecological responsibility. Green IT is one aspect of the relationship between ICT and sustainable development. Understanding the environmental impact of these technologies allows us to take better measures to reduce it. Awareness of environmental issues and the integration of information systems into the organization's environmental policies should encourage the development of environmentally friendly information and communication technologies.

Finally, although their number is still small, more and more companies around the world are integrating their green IT strategies into overall corporate social responsibility (CSR) projects. In this article, we will describe the main impacts of ICT on the environment. Then, we will present a solution proposed by digital responsibility and the benefits expected from the adoption of this technology.

## 2. Methodology

The narrative review is a means or means of conducting a review of the literature according to the question, topic or phenomenon of interest. It is also a tool to produce a scientific summary of the evidence in a particular field. In order to achieve the final result, there are six steps that need to be reviewed, some of them with iteration [5]. The steps illustrated in Figure 1 are important and serve as a guide when conducting the actual review.

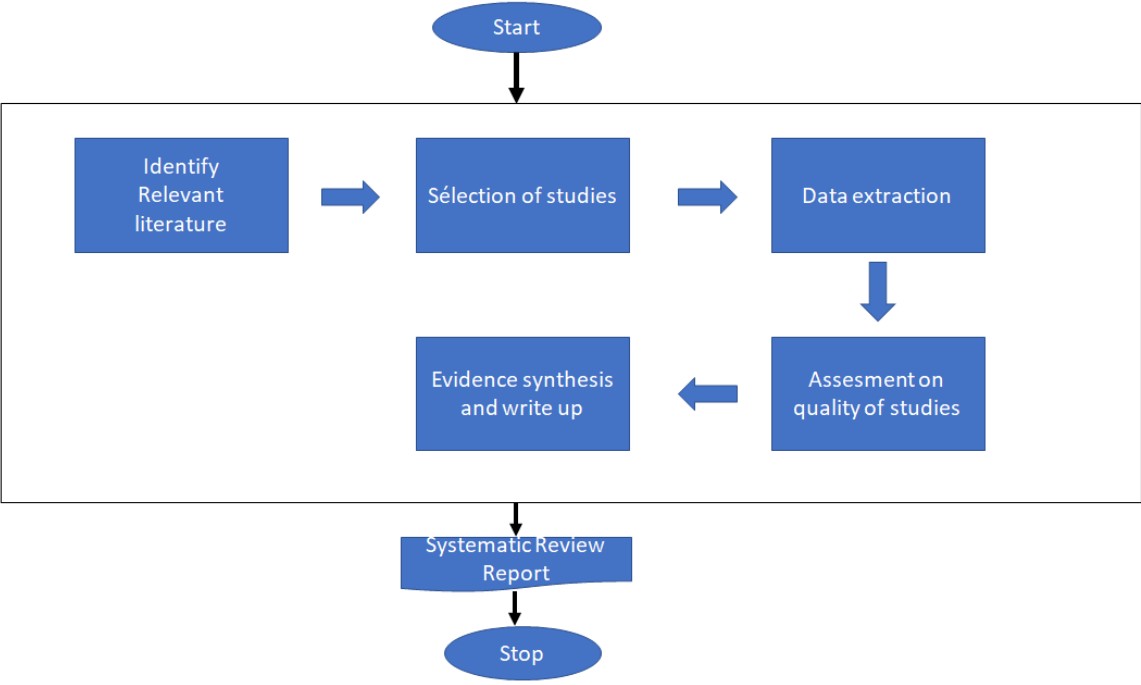

**Figure 1.** Stages in a Review Process.

### 2.1. Research Questions

The narrative review requires the formulation of research questions (RQs) that are used to guide the search and retrieval processes. The formulation of these RQs must include the first step of locating studies relevant to the research questions to be addressed and

identifying the search terms that will be used in the search process. These search terms can be considered the key elements that underlie the research questions.

These key search terms serve as the basis for deriving the relevant search terms that are used in the primary and secondary source research process. This paper reports on the first two primary research questions formulated. In order to identify and evaluate all available research on green computing practices, the following research questions are formulated:

- RQ1: What research has been conducted on green computing practices?

To answer this research question, this study aims to research existing research conducted on green computing practices that can benefit current and future research in this area.

- RQ2: What are the tools and future perspective that influence actors to implement green computing practices?

With respect to this research question, this study aims to identify the tools and future perspective that drive actors to implement green computing practices as a standard [6].

### 2.2. Conducting the Review

The first strategy for identifying relevant literature is to develop a search string that will be used in the search, this step involves the use of synonyms, alternative spellings and abbreviations. Once the search terms are identified, all the keywords will be compiled into a search string that will be used in the search process. The search can be performed using the Boolean operators OR and AND. The OR operator is used to group the various forms (e.g., synonyms and alternative spellings) of individual search terms. The AND operator is used to link all the different search terms into a single search string.

For the review to be considered reliable, the review process must be both transparent and, to some extent, reproducible. The set of primary sources typically comes from online databases, search engines, conference proceedings, peer reviews and journals, and all gray literature. The gray literature in this study considers material published from 1992 onwards, as the idea of green computing was born in 1992 with the launch of the Energy Star by the US Environmental Protection Agency (EPA) [7].

### 2.3. Inclusion and Exclusion Criteria

The literature selection went through five phases that are presented in Figure 2.

| Phase | Description |
|---|---|
| Phase 1 | Identify potentially relevant sources (from online databases or manual search) (n =130) |
| Phase 2 | Selection: Studies screened (title) (n = 130) |
| Phase 3 | Selection: Studies screened (abstract) (n =110) |
| Phase 4 | Selection: Studies screened (full text) (n =55) |
| Phase 5 | Studies included in the synthesis (n = 45) |

**Figure 2.** Inclusion Phase.

At the beginning of the study, 130 articles were selected. In the next phase, only 45 of these studies met the inclusion criteria. The function of the inclusion and exclusion criteria is to ensure that only relevant articles are selected for the review.

During the initial selection phase, all retrieved articles go through a filtering process. Screening consists of examining the title and abstract to identify relevant articles. An article that meets the minimum requirements for inclusion is selected. The articles are then

reviewed, and a decision is made to include or remove from the library. At this point, 110 articles have been selected from the 130 articles found in the database. The final part of the selection criteria involves reviewing all articles in detail. Articles are produced in hard copy and read. If an article did not meet the inclusion criteria, it was therefore discarded from the database. In the end, only 45 of 110 articles were selected. Figures 3 and 4 illustrates the approaches to the search and selection of related studies [6,7].

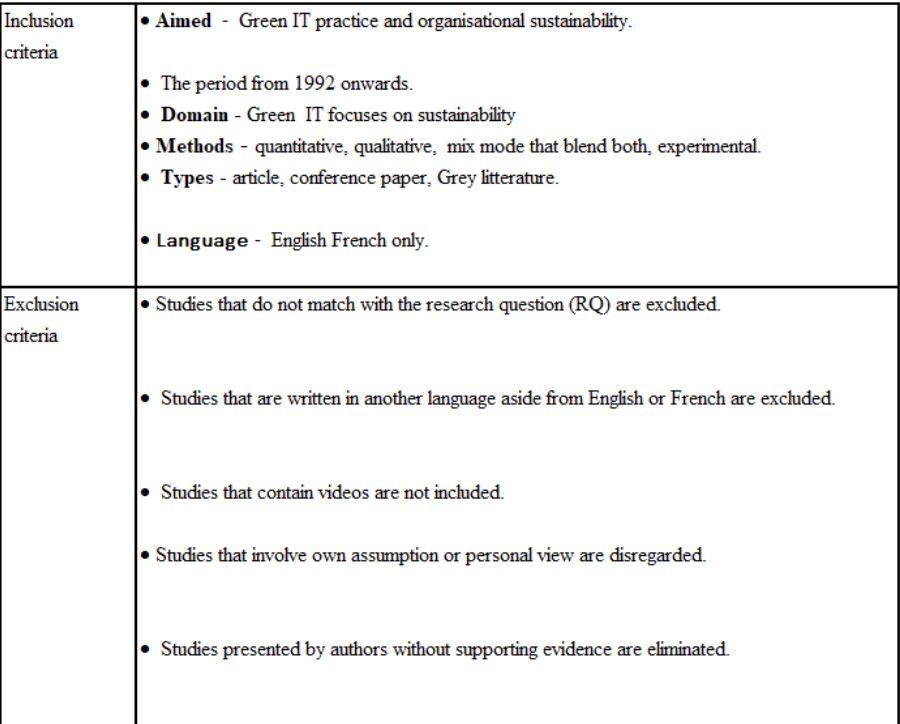

| Inclusion criteria | • **Aimed** - Green IT practice and organisational sustainability.<br><br>• The period from 1992 onwards.<br>• **Domain** - Green IT focuses on sustainability<br>• **Methods** - quantitative, qualitative, mix mode that blend both, experimental.<br>• **Types** - article, conference paper, Grey litterature.<br><br>• **Language** - English French only. |
|---|---|
| Exclusion criteria | • Studies that do not match with the research question (RQ) are excluded.<br><br>• Studies that are written in another language aside from English or French are excluded.<br><br>• Studies that contain videos are not included.<br><br>• Studies that involve own assumption or personal view are disregarded.<br><br>• Studies presented by authors without supporting evidence are eliminated. |

**Figure 3.** Inclusion and Exclusion Criteria.

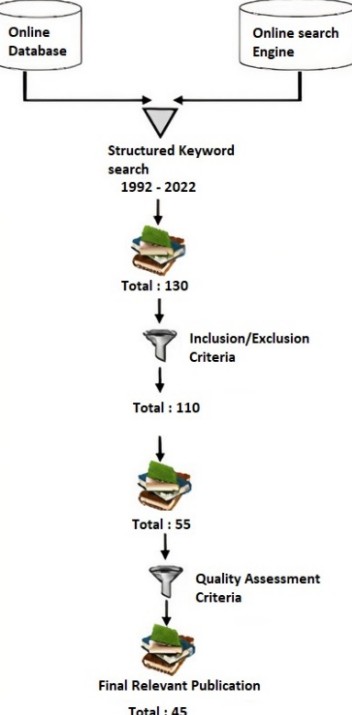

**Figure 4.** Stages in a Review Process.

## 3. State of the Art

In this section, we will define what responsible digital is.

### 3.1. Définition

Green computing (green information technology or responsible digital) is the practice of environmentally sustainable computing. Responsible digital aims to minimize the negative impact of computing operations on the environment by designing, manufacturing, using and disposing of computers and computing products in an environmentally responsible manner. The motivations behind green computing practices include reducing the use of hazardous materials, optimizing energy efficiency over the product's lifetime, and promoting the biodegradability of unused and obsolete products [6].

The concept of green computing emerged in 1992 when the U.S. Environmental Protection Agency launched Energy Star, a voluntary labeling program that helps organizations save money and reduce greenhouse gas emissions by identifying products with superior energy efficiency. Other components of green IT include data center redesign and the growing popularity of virtualization, green networks and cloud computing [7].

The concept of digital responsibility did not emerge overnight. It is the result of many initiatives such as non-governmental organizations, international organizations, governments, research centers and consulting firms [8]. To a large extent, NGOs (Non-Governmental Organizations) and associations have contributed to the debate around the topics of concern here, working with many experts, issuing regular alerts and publishing well-documented reports. These include the Responsible Digital Institute, Greenpeace, WWF and Friends of the Earth. For example, since 2006, Greenpeace has regularly compiled a constructive ranking of digital companies according to various criteria: energy efficiency, use of recycled products, use of non-renewable resources, such as minerals. The WWF is also very committed to the digital and sustainable development of construction sites. Its actions take the form of partnerships with companies and consumer awareness of eco-responsible use of digital technology [9].

In terms of research, these methods are more or less theoretical or experimental, and are associated with improvement or innovation processes. The main themes addressed are: metal, new materials, components, pollutants, recycling processes, ecological design, battery technology, global analysis (life cycle analysis—LCA, social life cycle analysis—SLCA, environmental assessment), connected objects, Internet objects, equipment intelligence and energy (measurement, modeling, optimization, renewable energies). The participants are mainly researchers from universities, IPCC, Euratom, electronic equipment manufacturers and private companies providing services in this field. They are participants who contribute to all the tools, databases, standards, etc... allowing research and development to progress.

The main challenge of digital responsibility is to make digital technology beneficial to both the environment and people. To achieve this ideal, the objectives are:

- Reduce the economic, social and environmental footprint of digital technology;
- Reduce the nuisances of the digital sector through its various stages: manufacture of computer equipment, use (energy consumption) and end of life (management and recovery of waste, pollution, depletion of non-renewable resources);
- Making digital technology accessible and ethical for all.

### 3.2. Key Concept

In the key concepts section, we will explain all the key concepts and definition of digital accountability by illustrating them with examples.

#### 3.2.1. LCA

First, life cycle assessment (LCA) is the only reliable method for determining the environmental footprint of a product or process [10]. This process involves the analysis of the material flow in and out of each stage (manufacturing, transportation, sale, use and end-of-life) of the product or process life cycle. For each stage, we calculate the incoming

flows—energy, electricity, raw materials, toxics, etc.—to arrive at the finished products, waste, greenhouse gases, etc. Since the scope of the research must be determined, the LCA is limited to direct effects. By repeating this method over the entire use case, it is possible to isolate the stages of the most polluting scenario.

For example, the LCA of a computer clearly shows that the most environmentally damaging steps are manufacturing and disposal. Manufacturing a computer in China emits 24 times more carbon dioxide in 1 year than using it in France. We also remind you that 8 out of 10 computers are still in landfills and have not been reprocessed [11]. Contrary to what the whole computer industry claims, the most effective posture to protect the environment is not to buy a new energy efficient computer, but to keep the existing one and use it as long as possible.

Consequently, the LCA methodology has been standardized by the various ISO standards. Standards:

- ISO 14041: Objective and scope of research;
- ISO 14041: Inventory;
- ISO 14042: Impact analysis;
- ISO 14043: Interpretation of results.

To carry out an LCA, there are different databases, depending on the industry. There are free databases, such as the IMPACTS database. However, most of them are charged. This is an additional cost to be passed on. For example, the cost of access to the ECOINVENT database is EUR 750 per user per year. Like any method, it has certain limitations that should be kept in mind. LCA is a tool, it takes into account many conditions, but not all existing conditions! Indeed, some aspects are not taken into account. For example: noise, odors, weather, light pollution, impact on the landscape, etc. LCA is therefore not the only eco-design method. It is the expression of a desire to design a product that respects the principles of sustainable development and the environment by using the least possible non-renewable resources. This method may not be appropriate in some cases.

In addition, the results depend on initial assumptions, which may be poorly calibrated or even wrong. Before beginning life cycle research, a series of questions should be considered. Since this method requires a great deal of time and effort, it must be carefully considered.

Illustration:

To illustrate an LCA, let us take the example of a smartphone. Throughout its life cycle (from raw material extraction to manufacturing, transportation, use and end of life), the smartphone will have an impact on the environment, society and health. The main impact of smartphones on the planet is the consumption of resources and the damage to biodiversity due to the emission of greenhouse gases into the environment. The manufacturing of smartphones (from mineral extraction to final assembly) accounts for about three quarters of these effects, mainly due to the screens and complex electronic components (microprocessors, etc.). The use phase of smartphones has a lower impact. The end of the life cycle depends on the recycling of the smartphone, which can have different effects [12].

In 2008, CODDE [13] analyzed the life cycle of an "average" second generation cell phone, revealing that:

- The production phase has the greatest impact;
- The transport phase has a very low impact on all environmental indicators, with the exception of ozone depletion where it generates 10% of the pollution;
- The use phase is responsible for 1% to 19% of the effects.

Then, through further research, the parts of the cell phone that have the most impact on the environment can be determined by:

- LCD screen;
- Electronic components, except for the battery and the screen;
- Lithium–ion battery;
- Charger.

Another study by Eric Williams, Jinglei Yua and Meiting Jua of Arizona State University and Nankai University in China analyzed the LCA and material flow of cell phones to calculate the overall footprint of cell phones manufactured and used in China. The results also highlight the manufacturing process, which concentrates all the energy consumed during the whole life cycle on 50%, against only 20% in the use phase. In contrast, for France, the weight of the use phase is about 5%.

Through LCA and environmental impact assessment, eco-design can identify areas for improvement, such as eliminating environmentally harmful chemicals or designing recyclable boxes [14]. It is also necessary to take into account the consumption of the device. Indeed, the use of the telephone during the last 3 years has only represented a small part of the "carbon dioxide" emissions in the world. In France, this share can be divided by 10, because the electricity is decarbonized thanks to nuclear power [15].

The energy consumption of a smartphone during its use depends on 4 factors:

- Battery capacity;
- Load frequency;
- The efficiency of the billing chain;
- Use of external batteries.

In about 50 samples, including the top 10 best-selling smartphones of 2017, observations on smartphone features show that:

- The battery capacity increases with the size of the screen;
- For the most popular smartphones, the battery supply voltage varies slightly (between 3.7 V and 3.85 V), ranging from 3.8 V to 3.82 V;
- The considerable difference in battery capacity associated with the corresponding autonomy, in most models, tries to maintain a battery life between 12 and 14 h (or 1.5 days of use).

Actual performance data for standardized USB power supplies are scarce, keeping the minimum required by the standard.

To illustrate this Figure 5, we use the graph above. The International Energy Agency offers a calculator that allows us to estimate $CO_2$ emissions depending on the device, the network and the country we are in. We used this one to create our graph. In gray, we have the $CO_2$ equivalent footprint of the construction of the device and in yellow the footprint of the consumption during its use. In green, we have added the consumption of the device compared to the average of the European electricity mix, and in blue, if it is connected to the French electricity network. A television set, for example, emits an average of 900 kg of $CO_2$ during its estimated 7-year lifespan. The share of consumption depends very much on the country in which one is located, and it is especially the construction that will emit the most $CO_2$ [16].

### 3.2.2. Rebound Effect

We could not define responsible digital use without talking about the rebound effect. In a very general way, the rebound effect, also known as Jevons' paradox, can be defined as "an increase in consumption associated with a reduction in restrictions to the use of technology, these restrictions can be monetary and temporal, social, physical, etc. The conclusion follows: the energy or resource savings initially anticipated by the application of the new technology are partially or totally offset by the adjustment in the company's behavior. This is of great importance for the formulation, evaluation and updating of energy strategies and policies [17].

Illustration:

A rebound effect of (for example) 10% means that 10% of the energy efficiency improvement initiated by the technology improvement is offset by the increase in consumption. The rebound effect can be expressed as a percentage of the energy efficiency improvement potential predicted by the engineer. The energy efficiency improvement should be measured in physical units.

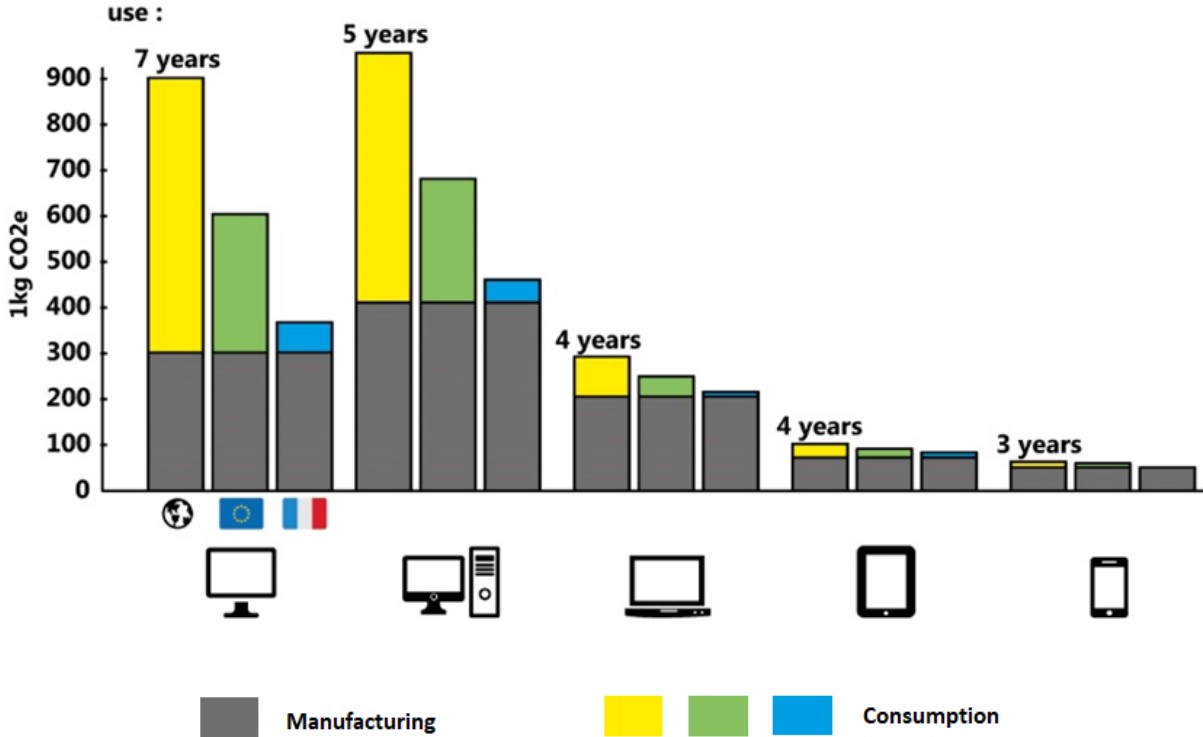

**Figure 5.** $CO_2$ by use.

Few studies have focused on rebound effects and ICT. Plepys (2002) has pointed out that the Internet can promote an increase in consumption (and thus have a rebound effect) because it can encourage the identification, comparison and purchase of goods and services. Thus, when a community provides its territory with a high-speed Internet connection, it can disseminate content that is favorable to sustainable development, but also increase tenfold the consumption (and production) of goods and services that consume materials and resources. In terms of energy for the moment, as a careful reading of the European Commission's communication (2008) calling for the diffusion of ICT to improve energy efficiency shows, the negative social and environmental impact of ICT is still ignored [18].

### 3.2.3. WEEE

On the other hand, many chemicals are linked to ICT (mercury, lead, cadmium, chromium, polybrominated diphenyl ether, etc.) that are dangerous for the environment and health. Especially when designing or destroying equipment at the end of its life, if workers are not sufficiently protected. This is often the case when e-waste is illegally shipped to emerging countries to be reconditioned. ICT waste is called Waste Electrical and Electronic Equipment (WEEE). The United Nations Environment Programme (UNEP) estimates that approximately 200 to 250 million tons of WEEE are generated worldwide each year, and that it is growing at a rate of 3 to 5 percent.

Illustration:

The lifespan of computers has been reduced from 6 years in 1997 to 2 years in 2005. In industrialized countries, the life span of cell phones is less than 2 years. This waste will lead to soil and air pollution. By infiltrating the soil and then the water table, the toxic substances go up the food chain, right onto our plates [19].

### 3.2.4. PUE

For the following, we need to define what PUE (Power Usage Effectiveness) is. PUE is the ratio of the total energy consumed by the data center to the total energy used by its IT equipment. According to the Uptime Institute in its 2019 report, the average PUE

of a private data center is defined as the end user or group cloud environment. In 2019, it was 1.67, which means that the total amount of 1 kW of data that the center's IT power consumes is 1.67. The power consumption is 1.67 kW [20].

Illustration:

Ideally, PUE should be close to 1 (just as some hyperscale data centers are close to 1.1). To achieve this goal, some companies have sought to develop new, low-power cooling methods. This is important because heat dissipation in today's data centers accounts for 40–50% of their power consumption (historically, PUE can be improved by installing hot and cold aisles).

### 3.2.5. Cloud HyperScale

It should be noted that the main environmental impact of the data center lies in the actions that humans can take directly: the first is their use of the data center, i.e., the way they use the internal IT infrastructure. Indeed, 90% of the greenhouse gases of data centers are emitted during the use phase.

Illustration:

Today, it is possible to optimize the utilization rate of the data center's IT infrastructure, which is defined as the ratio between the configured number and the actual number used. For example, in France, for some reasons, this ratio is in the order of 10 to 30% in certain data centers: peak loads that should require more infrastructure, regulatory restrictions in certain sectors such as banking, finance, healthcare, etc. It is impossible to consider a 100% utilization rate, but it is always possible to improve the utilization rate and thus reduce the environmental footprint of the data center [21].

## 4. Sectors and Solution

In the sectors and solutions part, we will come back to the three categories of the introduction, but this time we will deepen each of them with sources, key figures. Then illustrate in a sub-section the solutions that can be brought by responsible digital and all the corresponding techniques for each category.

### 4.1. DC

Computing requirements have exploded in recent years. Between 2010 and 2018, they have increased sixfold in data centers, while network traffic has increased tenfold and storage capacity has increased 25-fold. With such an increase, one might expect a proportional increase in energy consumption. However, this is not the case. Despite this strong growth, the power consumption of data centers barely increased between 2010 and 2018: about 6% according to J.G. Koomey and other scientists who published a recent study in Science [22].

At the same time, the majority of data centers designed in the last 10 years have implemented various techniques—containment, ASHRAE-compliant equipment, etc.—to reduce their cooling requirements and use free cooling. The PUE (Power Usage Effectiveness) of a recent data center has been cut in half in 20 years, from 3 to 1.5. In the end, up 6% between 2010 and 2018, data center electricity consumption is capped at 205 TWh per year (including 130 TWh for IT equipment), or about 1% of overall electricity consumption.

Solution

Heat dissipation in the data center is one of the major issues, but it is also one of the least known issues in the IT environment. Compared to previous equipment, new IT equipment that consumes as much or more power is becoming increasingly compact. As a result, the amount of heat generated in the data center has increased. Specific cooling and heat removal systems are used to collect and transport this excess heat energy for release to the atmosphere.

The first method is natural cooling: this involves locating data centers in cold locations (Nordic countries, underground, etc.) in order to use cool air to cool them naturally, thus

minimizing air conditioning [23]. For example, the American–Norwegian company Kolos chose to build a huge data center in the Arctic region of Ballangen in northern Norway. There are also many data center construction projects in Denmark, where the internet giants (Google, Facebook, Apple) are building their data hosting infrastructures.

Another trend is the Natick project launched by Microsoft in 2018, which involves the creation of an underwater data center in the Orkney Islands in Scotland. It houses 864 servers, all naturally cooled by seawater. Recently, a preliminary assessment of this experiment was conducted, which was very positive. In addition to using renewable energy for cooling, underwater data centers would experience only 12% of the normal failure rate observed on Earth, and would therefore be more reliable. On the other hand, the risk of corrosion is high and maintenance is indeed complicated [23].

However, both of these solutions are not feasible for all companies, because in some cases the data center must be located close to the user to ensure optimal performance, and also because some countries require that data be stored on the user's network. Another method, evaporative cooling, may be suitable for hot countries where data must be stored in the same territory. This technology moves warm air over a moist honeycomb cooling pad. The warm air then passes through a honeycomb radiator, which absorbs the heat and cools it naturally, hydrating the air. The cool air is then pushed into the atmosphere. This achieves the desired temperature without the need for mechanical cooling, reducing energy consumption and achieving a PUE of approximately 1088. However, this method has limitations because while it reduces energy consumption, it results in increased water consumption [24].

For data center cooling, the oil immersion method is also becoming increasingly popular. Here, instead of heating the air in the entire data center, the IT equipment is directly immersed in oil. Unlike water, oil has the advantage of not conducting electricity. In addition, with this method, the server can be protected from oxidation, chemical contaminants, moisture or thermal shock. Some data centers that are still applying this solution in the experimental stage have achieved a PUE of 1.04.

However, this technology will result in greater complexity in the maintenance process, will require more floor space than rack-based solutions, and may result in significant costs. This is because while there will be no additional costs when implementing this solution in a new data center, the installation cost, development cost, etc. will be different for the constructed data center. Finally, attention should be paid to the limitations of PUE-based power improvement perspectives. Indeed, as the Uptime Institute notes in its 2019 report, "data center energy efficiency improvements have leveled off or even deteriorated slightly over the past two years."

*4.2. Networks*

Greenhouse gas emissions from communications networks are at least as high as those from data centers. Between Google and our computers, there are many machines that consume a lot of power, and that power consumption will result in greenhouse gas emissions. In 2015, grid power consumption was estimated at 2142 terawatt hours. Wired networks such as ADSL or fiber accounted for about half, while the other half was wireless networks such as 3G or 4G. That could change, for the same amount of data exchange, with the use of 5G.

This growth is mainly related to the explosion of DSL/fiber TV and the dramatic increase in VoD, but not only [25]. The generalization of unlimited 4G packages is the other major culprit. According to our calculations, the weight of web pages has quadrupled in 10 years, from 450 KB in 2010 to 2000 KB in 2020.

Solution

To illustrate the solutions provided by responsible digital technology in the field of networks, let us take the example of video, which alone represents more than 70% of Internet traffic. Let us take a 20-min video. The International Energy Agency (IEA) [26]

offers a calculator that allows us to estimate the $CO_2$ emissions depending on the equipment, the network and the country we are in, considering only the energy consumption and not the construction of the equipment. Unfortunately, as we have seen on the user side device, it is clear that this structure emits the most greenhouse gases.

However, it gave us an idea. If we watch the high-definition video with a smartphone connected in 4G in France, the footprint will be 8 g of $CO_2$ equivalent, for example, if we use WIFI on the Internet box, it will be 2 g. If we watch this video on a 50-inch LED TV in WIFI, the use rises to almost 4 g, and for a laptop in WIFI, it is 2.4 g. Even without considering the data center structure, our video weighs only a few grams. Compared to driving a mile in a car, its emissions are relatively less. For example, it produces about 100 g of carbon dioxide [26]. Now, the same configuration, but in another country (e.g., Australia), we can see that the electrical structure of the country we are in is significant [25].

Very simply, the coefficient between the two is 10 in Figure 6 above, to which we have in Yellow the consumption of the device and in Green the consumption of the network. We can use these figures to explain that the impact of digital technology on the climate depends mainly on the structure of the devices we use and the quality of the electrical energy we consume, i.e., its low carbon content.

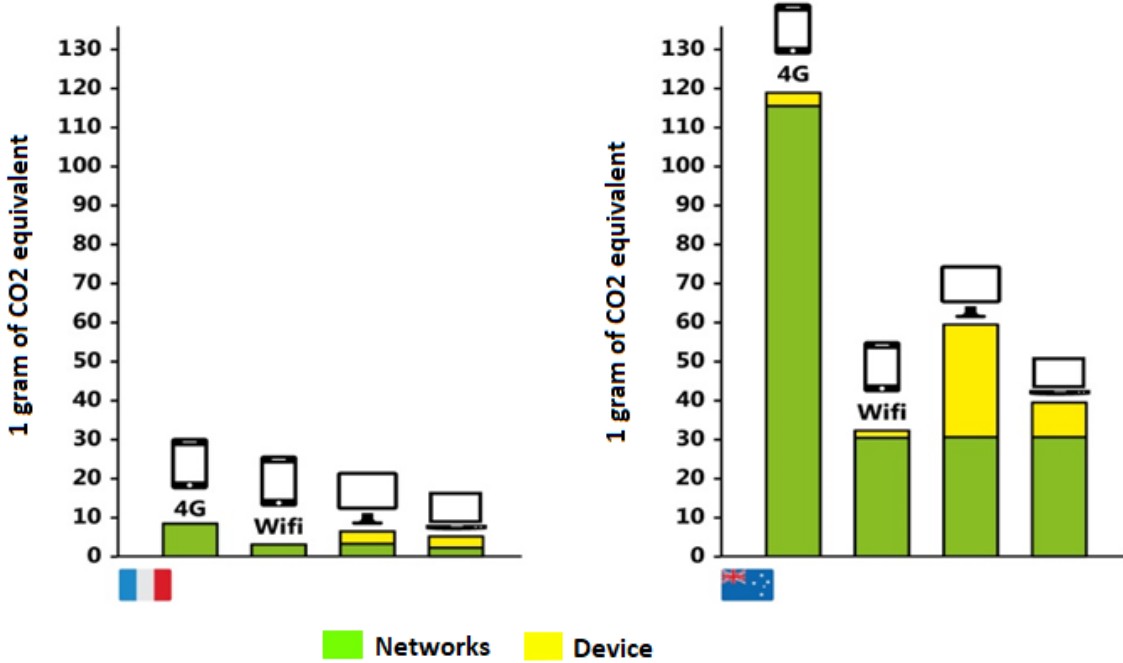

**Figure 6.** Consumption by location.

However, comparing France and Australia is not really easy, because not only the regulations in force and the communication frequencies are different, but also the number of inhabitants per km$^2$ (the population density is 3 in Australia against 119 in France, source: World Bank data). Therefore, systemic methods are needed to assess the environmental effects, such as the method proposed by CISCO [27].

*4.3. Equipment*

For equipment, some of the results of the LCA (life cycle assessment) study of stationary PCs show that, depending on the geographical location of the manufacturing process, the technology of the PC studied, the life span chosen and the manufacturing and transport assumptions, the results will be completely different [28,29]. Nevertheless, some trends emerge from these different studies:

Compared to the other stages of the distribution phase (transport), whether it is a short distance country or an international stage, its impact is negligible. The stages

of manufacturing and use of fixed PCs are those that have the greatest impact on the environment. During the manufacturing phase, the production of electronic components will bring the most harmful effects, and the assembly phases of the PC are less harmful for the environment. The use phase is directly related to the power consumption, and therefore more or less directly related to the "clean" power of the place concerned, If the recycling phase is carried out in accordance with existing rules (such as European standards), it can bring environmental benefits.

For cell phones, Eric Williams (Arizona State University, Tempe, AZ, USA), Jinglei Yua and Meiting Jua (Nankai University, Tianjin, China) conducted a study which showed that the manufacturing industry concentrates 50% of its energy consumption in the full life cycle and only 20% in the use phase and 30% in the recycling phase. In a country such as France, where 80% of electricity comes from nuclear power, the weight of the use phase is even lower (about 5%). The results of their research on laptops [30]:

- Use it as long as possible to avoid making new ones;
- Promote ecological design without chemicals that are harmful to the environment and users, and promote recycling.

Solution

Different standards, labels or tags can provide the most environmentally friendly equipment to different equipment buyers. Their choice depends on the criteria sought and determined upstream by the company. When selecting a supplier, the Information Systems Department (ISD) must first ensure that the product meets clear requirements, scalability related to software development, and even the availability of parts. With basic warranty or standard warranty extension.

Thus, the energy label focuses on the level of energy consumption of the product without taking into account other forms of impact (pollution, recyclability, etc.). The American Energy Star label is the best known. It is linked to the energy consumption during the use phase. This label is available for about 50 product categories in the United States, including washing machines, refrigerators and cell phones. In Europe, the label only applies to office equipment (computers, monitors, printers, copiers, fax machines, etc.) The current version is Energystar 8. Please note that the European Commission and the U.S. government have an agreement through the Environmental Protection Agency (EPA) to use "Energy Star" in Europe. This agreement expired on 20 February 2018. [31] On the other hand, environmental labels are ecolabels intended to cover the environmental impact of products, from raw material extraction to recycling (through use). Among the main labels for IT products are Nordic Swan, Blue Angel and TCO [32].

In addition to environmental standards, the so-called CSR (Corporate Social Responsibility) label also provides information on supplier governance, business ethics and social practices. Of the labels focused on IT products and services, only Epeat claims to take into account CSR standards, such as environmental and social responsibility performance standards in IT design. Epeat has become the most widely used label on computers and monitors in the U.S., expanding to printing equipment (copiers, printers, scanners...) since 2020, and more recently to cell phones.

## 5. Perspective

In this section, we will cover the outlook with studies and figures supporting the three main sectors as well as future sectors such as artificial intelligence and blockchain.

It is necessary to address the percentage of global emissions, which is the total equivalent of $CO_2$ we are sending into the atmosphere, all of which are based on current emissions. A study published in SUBSTAINBILITY estimates that emissions are expected to stagnate by 2020, when this article was published. This estimate is challenged by data from the SHIFT PROJECT, which estimates that GHG emissions will rise from 3% of global emissions in 2015 to 4% in 2020, and then to nearly 8% in 2025. Another study we use proposes two curve profiles for 2040. According to the authors [33,34], the most realistic is an exponential

curve, which can lead from 3% of emissions in 2020 to 14% of emissions in 2040 (see Figure 7). A percentage of 14% would mean that the digital sector would consume as much as the current agricultural sector, which is inconceivable if we want to meet our climate commitments [33].

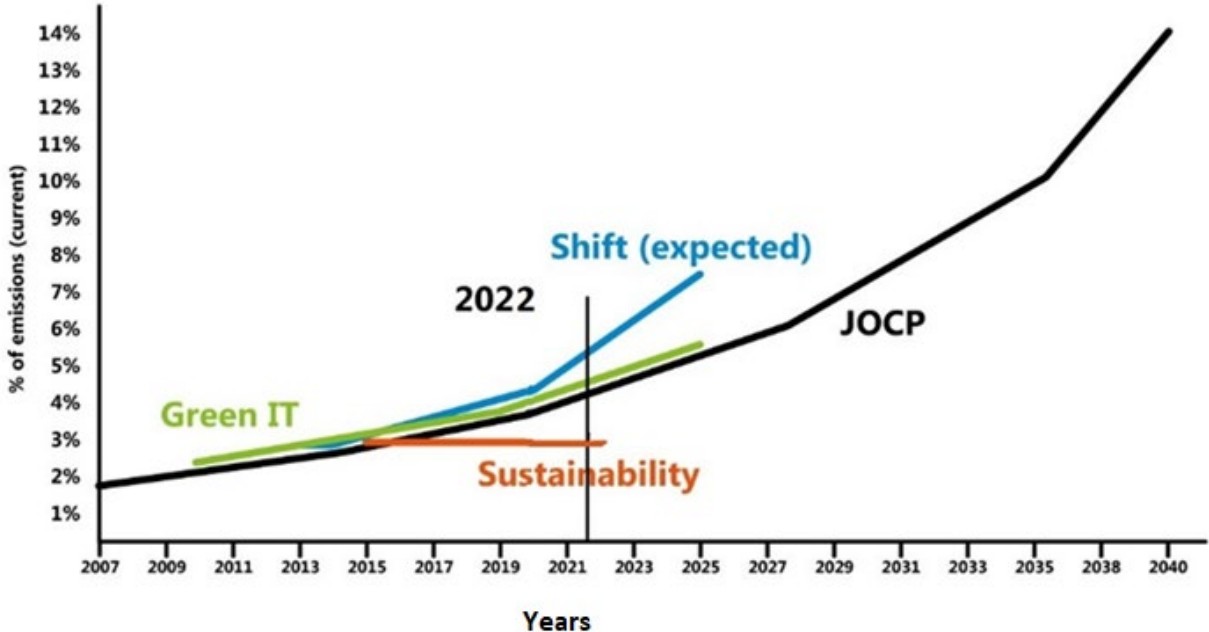

**Figure 7.** Future emission.

Thus, we will not address the resulting impacts, such as a shortage of certain minerals due to replacement. On the other hand, it is more likely that emissions in 2040 will be considered a trend, not a forecast, we will not necessarily reach that number because we, the authors of the research, are well aware that technology will evolve a lot in 2040. Indeed, while we can know what will happen in the labs in the next 5 years, it is complicated to know most of the applications of those same technologies. For example, Cisco estimates that Ultra HD VR streaming will account for the majority of the most used bandwidth in the future [35].

The most important impact is related to user equipment. Therefore, we need to pay attention to end-user equipment in home and business environments, including network-connected facilities. Although they account for the largest consumption, this is simply because their use is not optimized [27].

*5.1. DC*

In the collective unconscious, the main reason for greenhouse gases in the digital galaxy is the data center [36]. However, they only account for a quarter of global emissions and only 14% in France, most of the greenhouse gas emissions (47%) are due to consumer devices (computers, smartphones, etc.), while 28% of the emissions are attributable to network infrastructure [36]. In an article published in the journal Science, researchers say that between 2010 and 2018, the amount of computing in data centers increased fivefold, while energy consumption remained almost stable (it only increased by 6%). The explanation for these figures is the continuous improvement of energy efficiency in data centers, solutions that operators use to minimize facility consumption.

Indeed, in 2008, the industry adopted a green code of conduct, developed jointly with the European Union, making it a pioneer in the green transition [36]. In a report published by Jonathan G. Koomey in 2019, he found that data center usage has increased by 56% in 5 years, while overall ICT energy consumption in the US has increased by 36%. According

to the latest EINS report, ICT's share of electricity consumption increased from 4% in 2007 to 4.7% in 2012.

In other words, we are dealing with steady and sustained growth, but not the "exponential" growth that some fear, as illustrated by a quote from GreenTouch: "ICT consumes relatively little energy, but its use is exploding." This steady and sustained increase in ICT's share of overall electricity consumption is paradoxical in itself, as needs are exploding and will grow very rapidly over the next 10 years if measured in terms of mobile and fixed traffic. It is important to understand what is driving this consumption up.

### 5.2. Networks

Of the three levels—users, networks, data centers—the network share increases the most in relative value. This is because the share of users is decreasing from −5% to −13% in 15 years (depending on the indicator observed), which proportionally increases the impact of the network and data centers. Another complementary explanation is that the deployment of at least 10 million 4G and 5G base stations (radio base station) between 2010 and 2025 is added to the impact of existing network infrastructure, especially DSL/fiber boxes with +10 million base stations from 2010 to 2025 [37].

### 5.3. Equipment

To illustrate the prospects for improvement in terms of equipment consumption and manufacturing, we can take the example of evaluating the energy consumption of Apple computers during their use phase. To meet the requirements of the computer specifications, we use the Typical Energy Consumption (TEC) Energy Star 8.0. Our annual energy consumption for the 2010 model was 49.11 kWh and our annual energy consumption for the 2011 model was 45.03 kWh. Both of our estimates are consistent with the Computer Energy Star 8.0 technical standards, which state that a computer's energy consumption should not exceed 53 kWh and should be certified annually [38].

## 6. Other Perspective

In addition to three tiers, the future challenges of responsible digital technologies in future areas such as Artificial Intelligence or IoT should be addressed to provide insights.

### 6.1. IOT

The Internet of Things (IoT) is a network of interconnected physical objects called the Internet of Things in French. These objects are integrated with specific sensors, software protocols and network adapters to enable the interconnection/linking of objects. Due to their unique identifiers, these objects can be associated with each other and collect, transmit and exchange data over the Internet without manual intervention. Therefore, the Internet of Things can remotely detect and control devices/objects on the Internet via the existing network infrastructure. Therefore, the Internet of Things has great potential to improve the efficiency, accuracy and financial benefits of almost all areas of life [39].

Currently, in 2020, connected objects include everything from homes, connected cars, connected medical institutions and everything used for inventory and material management. According to Cisco's forecast [40], by 2022, IoT connections will account for more than half of all connected devices and connections worldwide. By 2022, traffic generated by IoT-enabled devices is expected to reach 6% of global IP traffic.

### 6.2. 4G/5G Telecoms

Following 4G, 5G is the fifth generation of digital data transmission technology using electromagnetic waves. The wavelength used initially will be 3.6 GHz and will reach 26 GHz in a few years. Thus, the main environmental argument that justifies the deployment of 5G is that it consumes less energy. In fact, electricity is not an environmental indicator. If we use low-impact primary energy sources (such as dams on rivers) to produce large amounts of electricity, our environmental impact will be minimal. Conversely, if we

produce electricity from impactful resources such as coal, it will have a significant impact on the environment [41].

Thus, the popularity of 5G will inevitably shorten the lifespan of 2G/3G and 4G smartphones. Further, 5G also sells new multimedia uses that never existed or were previously impossible: high-definition video streaming, video game streaming, etc. on the go. Even if the industry denies it, people might be surprised by the cannibalization of wireline networks (DSL, cable and fiber) by radio wave networks (i.e., 5G). When all you have to do is put the antenna on the roof to offer faster speeds than ADSL, everywhere, even in white areas, what benefits can operators bring to digging trenches all over France? For this, operators are highlighting the energy efficiency gains possible with 5G. According to Orange, "to move 1 GB of data, 5G will use 2 times less energy than 4G at launch, 10 times less energy by 2025, 20 times less energy by 2030." In particular, "MIMO" smart antennas will improve the energy efficiency of 5G: these antennas will direct the radio signal to users soliciting the network, and will not be streamed continuously in all directions, for example as in previous generations of cell phones [42].

Compared to 4G, 5G will multiply the speed by 10. As a result, the increase in usage seems to compensate for, or even exceed, the increase in efficiency. Bouygues Telecom has confirmed the existence of this rebound effect: its CEO even admitted to the Committee on Land Use Planning and Sustainable Development that "after the first year of deployment, the energy consumption of all operators will increase significantly. "These remarks echo the opinion of the Shift project. In the Shift Project, "with 5G deployment, mobile operators' energy consumption will increase by 2.5 to 3 in the next 5 years." However, this estimate based on global research in 2019 does not come from a true country-level assessment.

### 6.3. Block Chain and Cryptocurrency

In the OPECST report n°1092, the author proposed the following definition of blockchain technology: "what is called by metonymy blockchain designates the technologies of storage and transmission of information, allowing the constitution of replicated and distributed registers (distributed ledgers), without a central organ of control, secured thanks to cryptography, and structured by blocks linked between them at regular intervals of time.

Bitcoin [43,44] is the first public blockchain network and remains the leading cryptocurrency dominating the market today. Several studies have attempted to quantify Bitcoin's energy consumption or more generally its environmental impact, and at least two websites offer graphical visualizations of this energy consumption over time. The estimation methods vary: the best one takes into account the energy mix of the countries/regions where the miners are located, but all give a comparable order of magnitude: the power consumption of the Bitcoin network in 2019 is 30 to 80 TWh per year. Between the two, the carbon footprint is 15–40 MtCO$_2$-eq, which is comparable to countries such as Austria, Belgium or Denmark [44].

The excessive energy consumption of some public blockchain technologies such as Bitcoin or Ethereum is not a fatality. This consumption of the order of small European countries comes from the consensus algorithm used: Proof of Work. Since the advent of Bitcoin in 2008, a new consensus algorithm has been proposed to offer another compromise between security, decentralization and energy consumption. As Ethereum is expected to migrate to Proof of Stake in 2020, this is a very important step towards a more energy-efficient blockchain technology. For all that, the energy consumption of blockchain technologies other than bitcoin is still under-researched, and it should be studied in depth and compared to the energy consumption of traditional distributed databases to measure the energy cost of not having a trusted third party [44].

### 6.4. AI

Artificial intelligence is a two-faced technology. It has been criticized for the high energy consumption required to run it. On the other hand, it is widely praised for the optimizations it can generate. On the other hand, artificial intelligence is an excellent

opportunity to better understand and solve environmental problems, but the drawback is that the computing power it involves is too energy intensive. New artificial intelligence in the green market tends to specialize along the seven broad themes we have identified: smart cities, energy conservation, connected agriculture, climate change modeling, environmental protection, sustainable mobility, and local and sustainable economies [45].

Artificial intelligence abuses all digital technologies: servers, networks, data storage. As a result, it significantly worsens the depletion of scarce natural resources and the continuous increase of global electricity consumption. The environmental impact of online services using artificial intelligence must be assessed as a whole. There is still a long way to go to realize the dream of green artificial intelligence, but taking inspiration from our own sober brains, the work of gradually empowering artificial intelligence is already underway. Bionics has always been an important driver of artificial intelligence development: a better understanding of our brains has enabled the development of artificial intelligence. With a power consumption of only 12.6 watts, the human brain shows us the way to improve the efficiency of AI methods [45].

## 7. Conclusions

In summary, the energy bill for ICT remains high and is expected to continue to grow as applications expand and the user base increases, despite the continued increase in IT energy efficiency (Moore's Law, standby and dynamic wake-up equipment). Energy therefore has a significant economic impact on telecommunications and data centers. In the latter case, performance increases have already been achieved and will continue. Much work remains to be conducted in telecommunications as well as in the sectors used today, especially since the expansion of cellular networks to 5G will require base station density to provide effective coverage in environments such as large cities with many obstacles to propagation of radio waves [5].

Although this study followed the methodology suggested in the Methods section, it has some limitations. The selection of keywords and the inclusion and exclusion criteria were based on the judgment of the researchers. Another limitation is that all of the selected literature was obtained from well-known databases that allow access to only limited articles. Inaccessible items will be automatically dropped. Therefore, to improve future research, this study recommends the inclusion of all databases. In addition, all selected literature was identified and judged on the basis of the researchers' knowledge and guidelines followed. Although consensus meetings have been established, there is still an associated risk of researcher bias, which must be minimized when evaluating the contribution of each article [5].

Based on a critical review of the gray literature in the field of green IT, green computing practices appear to require further research. While green IT practices are the focus of many concerns, organizational awareness of their implementation is low. Nearly half of the literature reviewed was on the environment. Therefore, as a first review for future research, this article proposes to focus on other areas rather than the environment as research topics and contexts. Organizations may be a good example of a domain to focus on when determining the contribution of green IT practices to the system itself. When an organization has a positive view of the practice, it is more concerned with energy consumption and environmental impact. Based on these results, more focus should be placed on the contributions and metrics of green IT in any area [46].

For companies, there are many issues at stake when adopting a responsible and sustainable digital strategy. As digital technology has an ever-increasing carbon footprint, rethinking the way we use digital tools is crucial for companies. The process often starts with the creation of awareness campaigns for the staff in order to develop good practices internally.

First of all, from an economic point of view: not only the optimization brought by Green IT will reduce energy needs but also the cost of ownership (TCO) of companies. Then, from an environmental point of view, Green IT is a must. Whether through legislation (mandatory reporting of corporate social responsibility, carbon footprint, carbon tax . . . )

or through a voluntary sustainable development policy, ICT are important levers for improvement [46].

Green IT, as well as a vector for reducing environmental and economic impact, is therefore a competitive advantage for our management methods: an investment for the future. Digital responsibility is therefore a cross-cutting issue that impacts different types of activities in the company. Thus, the design and implementation of the responsible digital approach requires the mobilization of the company's employees (or service providers) responsible for:

- Hardware and software design;
- their development;
- and the purchasing sector.

  Yet, also:

- end-users (business services, support functions, also called "Internal Customers" by the IOC);
- and employees responsible for waste management.

In order to better understand the environmental issues related to digital technology and the opportunities for action, it may also be useful to involve employees, suppliers, customers and representatives of environmental NGOs in this work. Like any horizontal approach, digital responsibility is a long-term process and its success depends on its adoption by politicians and end users. It must be supported by companies, which will dedicate human and financial resources to it by appointing, for example, green IT missions to oversee and coordinate actions.

To be more precise, our greatest leverage to fight against climate change at the digital level is to extend the life span of equipment as much as possible: "The best waste is the one we don't create". Finally, new themes are emerging and are part of the responsible digital framework, such as artificial intelligence and blockchain. So, many topics that are at the heart of digital issues and will shape the digital environment of tomorrow, but if we are only guessing here, one thing is certain: Green IT is the rising trend.

**Author Contributions:** Conceptualization, D.B. and C.V. and B.G.; methodology, D.B. and C.V. and B.G.; validation, D.B. and C.V. and B.G.; formal analysis D.B. and C.V. and B.G.; investigation, D.B. and C.V. and B.G.; resources, D.B. and C.V. and B.G.; data curation, D.B. and C.V. and B.G.; writing—original, D.B. and C.V. and B.G.; draft preparation, D.B. and C.V. and B.G.; writing—review and editing, D.B. and C.V. and B.G.; project administration, C.V. All authors have read and agreed to the published version of the manuscript.

**Funding:** This research received no external funding.

**Institutional Review Board Statement:** Not applicable.

**Informed Consent Statement:** Not applicable.

**Data Availability Statement:** Not applicable.

**Conflicts of Interest:** The authors declare no conflict of interest.

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
