# Peer review of "Review of the Impact of IT on the Environment and Solution with a Detailed Assessment of the Associated Gray Literature"

_sustainability, doi:10.3390/su14042457_

Round 1

Reviewer 1 Report

the paper does not have any specific hypothesis research questions. It has little scientific structure. The reports and perspectives used are very general and the authors' contribution is negligible. There is really no comparative analysis of the research to date. The only plus is an interesting topic.

Author Response

Dear Reviewer,

You will find in the Word file all the answers to your comments.

Sincerely,

Guillaume Bourgeois

Reviewer 2 Report

The title suggests for a critical topic of discussion. However, the content is not addressing the main aim of the paper and it was loosely written. What is actually the uptake of the paper? The content is mainly on the reviews of the methods related to green computing, green ICT, responsible digital and ecological responsibility, with no specific direction and a clear structure. In addition, the graphs were ambiguous, and have no indication of the information either on the axes, or the different color. The citations were not arranged accordingly, and the references deviates the formatting style. How does the paper address sustainable IT was not clearly discussed.

Author Response

(The authors gave the same response as above.)

Reviewer 3 Report

The manuscript joins the research efforts on promoting and increasing awareness of the environmental effects of information and communication technologies (ICT). In the ICT eco-system, the article refers to data centrescommunication networks and equipment and presents (in a scarce manner) the current technologies aimed at reducing the environmental footprint. Overall, the manuscript is difficult to read, lacks clarity and does not do justice to the rich scientific literature already on the topic. I would suggest to the authors to organize the content of the manuscript around a clear research question.  Related to this, there’s no Methodology section thus we have no information on how the review what performed, and about their empirical material. As a matter of fact, the entire manuscript relies on grey literature and even notorious papers (such as Masanet, E., Shehabi, A., Lei, N., Smith, S., & Koomey, J. (2020). Recalibrating global data center energy-use estimates. Sciencehttps://doi.org/10.1126/science.aba3758) are cited via grey literature (see reference 28). If this is a grey literature review, it should be clearly stated, and the choice motivated in the Materials and Methods section (or Methodology). As-is Materials and Methods section, presenting the current standards for the evaluation of environmental impact should be included in your State-of-the-art section. Additionally, the Abstract does not reflect the content of the manuscript and should be adapted accordingly.   

Overall, I support the urgency of addressing this topic, but this manuscript is too raw to envision its publication at this point. I would recommend the authors to revise the entire manuscript, identify the gap they are addressing and thus straighten their message, check the coherence of some ideas (regarding artificial intelligence and blockchain). I would advise the authors to carefully read their manuscript and correct punctuation errors (check Keywords, lines 237, 456, 467, 533 etc. ), adopt English punctuation, typing errors (pike line 458, capitalize CO2...), group sentences in paragraphs.  Finally, I will list below some of the observations that emerged when reading the manuscript (keep in mind the list is not exhaustive): 

Line 67: NGO acronym to be defined 

Line 84: What do they do, these participants?  

Line 113: There’s no “I” in teamwork. 

Line 128: Where do these numbers come from? Add references! 

Line 190: Is it an assumption or a fact that In France you can devise the emissions by ten because it’s nuclear?  

Figure 1: Where do these values come from? Under which assumptions are these valid? What is the European electricity mix? 

Line 250: The 2-year life span if given by whom? Technological obsolescence? Programmed obsolescence?  Add references! 

Line 293: The same wording appears twice. 

Line 320: Use land instead of Earth... 

Figure 2. How are these consumptions estimated? Comparing France with Australia is not really straightforward since not only have different regulations in place, communication frequencies but also a different number of habitants per square km ( the population density is 3 in Australia compared to 119 in France, source: a Google search) thus we would expect also less connected devices.  Following this logic, I would argue that the cumulative effect is more important than the individual one. Let's say for instance that all the population in a square km watch the same video on the mobile phone in France and Australia, then under the hypothesis used here, we would have a consumption (footprint?) of 952 (are these CO2 eq??) for France and approx. 360 for Australia for all mobile devices connected on a square Km. The message here is that you need systemic methods to evaluate environmental effects.  

What does the Y-axis show? CO2 eq? KWh? Is it the footprint of consumption?  

Line 506: Who’s lifetime are we talking about? Is it the user? 82 years? 

Line 511: The authors take the fixed emissions (manufacturing, transportation and end-of-life) and divide them to one-year use emissions. Should the fixed emissions be divided by the overall useful life emissions?  

Line 537: The idea emerging from the paragraph is not clear. 

Check references and their order!

Author Response

(The authors gave the same response as above.)

Round 2

Reviewer 1 Report

ArtykuÅ‚ zostaÅ‚ poprawiony, ale nadal zawiera poważne bÅ‚Ä™dy, nie ma konkretnych hipotez, problemów badawczych ani celów prowadzenia tych badaÅ„. Analiza literatury nadal nie jest krytycznÄ… analizÄ… dotychczasowych badaÅ„, a jedynie opisem tego, co inni zawarli w pracach o podobnej tematyce. Luka badawcza, jaka powinna wynikać z literatury, nie zostaÅ‚a sprecyzowana. Sam test jest przeprowadzany poprawnie.

Author Response

Dear reviewers,

We would like to thank you again for all your constructive comments.

You will find the answer in the attached word document.

Sincerely,

Guillaume Bourgeois

Reviewer 3 Report

After going through your revised manuscript, I still have some remarks:

  1. I think is important to mention that the paper presents some solutions already implemented in the field to reduce the effects of technology on the natural environment.
  2. What's the research gap you are addressing? Why a grey literature review?
  3. A sound research methodology typically includes the initial set of keywords used to extract the articles and clearly mentions the exclusion criteria. 
  4. I still think is important to mention the limitations of the results shown in Figure 2, i.e. comparing France to Australia.
  5. The Conclusions section does not mention any research limitations as you state in your Abstract.
  6. References are still missing, abbreviations, punctuations...   

Author Response

Dear reviewers,

We would like to thank you for all your relevant comments since the beginning which allowed us to greatly improve our manuscript. You can find in attachment the answers to your last report.

Sincerely,

Mr Guillaume Bourgeois on the behalf of the authors

Round 3

Reviewer 1 Report

Corrected according to comments.